# One-to-many Approach for Improving Super-Resolution

## Abstract

Super-resolution (SR) is a one-to-many task with multiple possible solutions. However, previous works were not concerned about this characteristic. For a one-to-many pipeline, the generator should be able to generate multiple estimates of the reconstruction, and not be penalized for generating similar and equally realistic images. To achieve this, we propose adding weighted pixel-wise noise after every Residual-in-Residual Dense Block (RRDB) to enable the generator to generate various images. We modify the strict content loss to not penalize the stochastic variation in reconstructed images as long as it has consistent content. Additionally, we observe that there are out-of-focus regions in the DIV2K, DIV8K datasets that provide unhelpful guidelines. We filter blurry regions in the training data using the method of [10]. Finally, we modify the discriminator to receive the low-resolution image as a reference image along with the target image to provide better feedback to the generator. Using our proposed methods, we were able to improve the performance of ESRGAN in $\times 4$ perceptual SR and achieve the state-of-the-art LPIPS score in $\times 16$ perceptual extreme SR.

## 1 Introduction

Super-resolution is the task of recovering a high-resolution (HR) image from a low-resolution (LR) image. Recent works have achieved significant performance in SR using deep convolutional neural network (CNN) based approaches. Some of them exploit strict content loss as the training objective for super-resolution and propose various network architectures to improve the PSNR score. However, these methods often result in overly smooth images and have poor perceptual quality [6]. Another branch of works focuses on improving perceptual quality with perceptual training methods [1,6,7]. These methods employ generative adversarial networks (GAN) and perceptual loss functions to drive the network's output towards the natural image manifold of possible HR images. We assess an1d further improve the perceptual quality of these works.

Because super-resolution is a one-to-many problem with multiple possible reconstructions for one image, methods based on strict content loss often lead to predicting the average of possible reconstructions[6]. Perceptual-driven solutions utilize perceptual and adversarial loss, which both don't penalize the generator for generating equally realistic images with stochastic variance. However, we discover two incomplete aspects in the current perceptual SR pipeline. First, although the above-mentioned losses don't penalize stochastic variation, the final loss is mixed with the strict content loss which strictly penalizes these variations. Second, the generator doesn't have the ability to generate multiple estimates of the image despite a one-to-many problem. To implement such a one-to-many pipeline, we provide the generator with pixel-wise noise and improve the content loss so it doesn't restrict the variation in the image while ensuring the consistency of the content.

The key contributions of our work can be described as follows:

Submitted to 35th Conference on Neural Information Processing Systems (NeurIPS 2021). Do not distribute.

- We propose a weaker content loss that does not penalize generating high-frequency detail and stochastic variation in the image.
- We enable the generator to generate diverse outputs by adding scaled pixel-wise noise after each RRDB block.
- We filter blurry regions in the training data using Laplacian activation[10].
- We additionally provide the LR image to the discriminator to give better gradient feedback to the generator.

## 2  Related work

Since the pioneering work of SRCNN[9], many works have exploited the pixel-wise loss and PSNR-oriented training objectives to learn the end-to-end mapping from LR to HR images. We denote such pixel-wise losses as the strict content loss. Many network architectures and techniques were experimented with to improve the complexity of such networks. Deeper network architectures[17], residual networks[6], channel attention[18], and techniques to remove batch normalization[19] were introduced. Although these works achieved state-of-the-art SR performance in the peak signal-to-noise ratio (PSNR) metric, they often produce overly smooth images.

To improve the perceptual image quality of SR, SRGAN [6] proposes perceptual loss and GAN-based training. The perceptual loss is measured using intermediate activations of the VGG-19 network and a discriminator is used for the adversarial training process. Enhanced SRGAN (ESRGAN) further improves SRGAN by modifying the generator architecture with Residual in Residual Dense Block (RRDB), the Relativistic GAN [16] loss, and improving the perceptual loss. Such methods were superior to PSNR-oriented methods at generating photo-realistic SR images with sharp details, achieving high perceptual scores. However, we could still often find unpleasant artifacts and problematic textures in the reconstructions of ESRGAN. Such cases are exemplified in Figure 4.

Traditional metrics for assessing image quality such as PSNR and SSIM (Structural Similarity Index Measure) fail to coincide with human perception[4]. The PSNR score is calculated based on the pixel-wise MSE, so methods that minimize pixel-wise differences tend to achieve high PSNR scores [9]. However, the PSNR-oriented solutions fail at generating high-frequency details and often drive the reconstruction towards the average of possible solutions, producing overly smooth images[6]. The learned perceptual image patch similarity (LPIPS) score[4] was proposed to measure the perceptual quality on various computer vision tasks. According to [2], the LPIPS score reliably coincides with human perception for assessing super-resolved images. We use the LPIPS score as an indicator of perceptual image quality in our experiments.

CycleGAN[8] is a pipeline for image-to-image translation with unpaired images using generative adversarial nets and cycle loss. CycleGAN consists of 2 generators $G_1, G_2$, and 2 discriminators $D_1, D_2$, where $G_1$ and $G_2$ each translate the input image in a cycling manner. The generators are trained to minimize the adversarial loss and cycle loss $||G_2(G_1(x)) - x||_1$ between the input image and cycled image. We were able to design a loss based on the cycle loss to reliably measure the content consistency without such a complicated design.

## 3  Method

We design a one-to-many approach for perceptual super-resolution by modifying the generator and the training objective. We also describe additional modifications to the training process and discriminator to improve the perceptual quality of SR.

### 3.1  Cycle consistency loss

Most works on perceptual super-resolution[1, 6, 7] combine the content loss, adversarial loss (GAN loss), and perceptual loss for the training objective as in Equation 1. Although the strict content loss and adversarial loss are fundamentally disagreeing objectives, relying exclusively on either loss each has significant issues. The strict content loss guides the network output to be exactly consistent with the HR image, guiding the network to learn the mean of possible reconstructions and thus tends to give overly-smoothed results. Although the GAN framework is a powerful method for

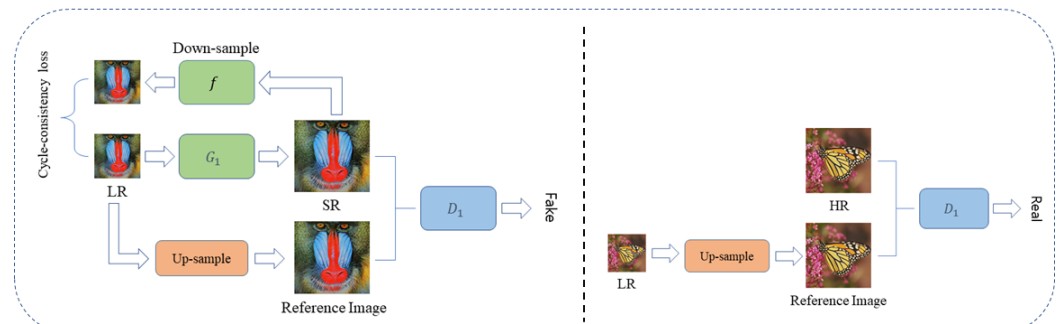

Figure 1: An overview of our method. The cycle consistency loss is measured by comparing the LR image with the downsampled SR image. The discriminator is provided with the target image and a reference image generated by bicubic-upsampling the LR image.

photo-realistic image generation, adversarial learning is highly unstable, and while the adversarial loss and perceptual loss guide the network to be perceptually convincing, they don't enforce the content of the super-resolved image to be consistent with the low-resolution image.

$$L_{Total} = L_{percep} + \lambda L_{GAN} + \eta L_1 \tag{1}$$

We regard simply trading off these disagreeing losses as an incomplete objective for super-resolution since the mixing of such losses will obstruct the optimization of either loss. An improved training objective must be GAN-oriented while ensuring consistent content of the image. That is, there needs a content loss that doesn't hamper the generation of images with high-frequency details.

We propose a soft content loss inspired by the cycle loss of CycleGAN[8] to ensure the output of the generator to be consistent with the low-resolution image while not disturbing the generation of high-frequency information.

We view the super-resolution problem as an image-to-image translation task between the LR and HR image space and apply the CycleGAN framework. To simplify the problem, we exploit our prior knowledge on $G_2 : HR- > LR$. We can denote the downsampling operation as $f$ and set $G_2$ to be $f$ instead of learning it. Consequently, our pipeline doesn't require learning $D_2$ which is a tool for learning $G_2$. This leaves only $G_1$ and $D_1$ to be learned. We can write the cycle consistency loss as Equation 2. This loss won't penalize generating high-frequency details in any way while the SR image remains consistent with the LR image. Finally, we can conclude our generator loss as Equation 3.

$$L_{cyc}(G_1) = ||f(G_1(LR)) - LR||_1 \tag{2}$$

$$L_{Total}(G_1) = L_{cyc}(G_1) + \lambda L_{GAN}(G_1, D_1) + \eta L_{percep} \tag{3}$$

### 3.2 Providing scaled Gaussian noise to the generator

For the generator to be capable of generating more than one solution given a single image, it must receive and apply random information. The variation between super-resolved images will mostly be stochastic variation in high-frequency textures. StyleGAN[3] achieves stochastic variation in images by adding pixel-wise Gaussian noise to the output of each layer in the generator. We adopt this method and add the noise after every RRDB layer in the generator.

However, the sensitivity and the desired magnitude of noise would differ for each channel. Adding the same noise directly after every layer could rather harm the ability of the generator. For example, a channel that detects edges would be seriously harmed by the noise. The sensitivity will also depend on the depth of the network. To mitigate such possible issues, we allow each channel to learn the desired magnitude of the noise. Specifically, before adding the noise to the output of each layer, we multiply

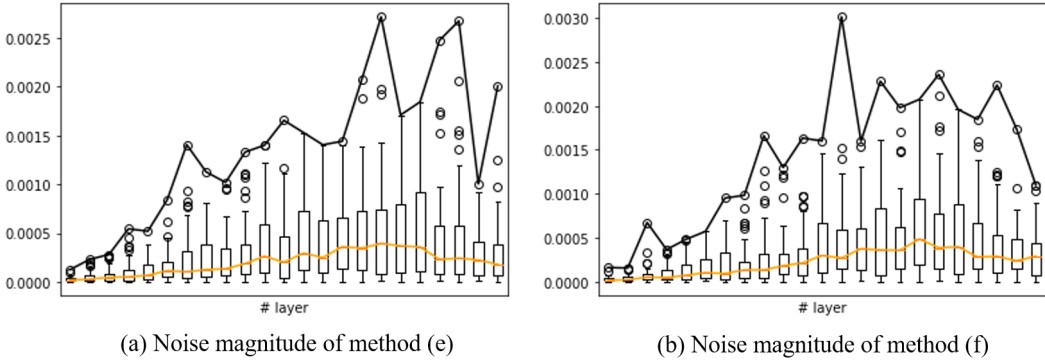

(a) Noise magnitude of method (e)  (b) Noise magnitude of method (f)

Figure 2: Boxplot of the scaling factors against the position of the layer in the network. The desired magnitude of noise increases in deeper layers, while the final layers have smaller scaling factors. The sensitivity to random noise varies for each layer and channel.

the noise with a channel-wise scaling factor. The scaling factor is learned concurrently with the network parameters. We observe that the desired magnitude differs along the network depending on the position of the layer. This shows that our method effectively implements a one-to-many generator for super-resolution. The early layers seem to be focusing more on extracting the feature of the image, while the final layers preferred the noise to be scaled before being applied to the reconstruction. Details are illustrated in Figure 2. The noise is not applied at evaluation.

### 3.3 Reference image for the discriminator

Traditionally, the discriminator network receives a single image and is trained to classify whether the given image is real or a generated image. This setting will provide the generator with gradients to "any natural image" instead of towards the corresponding HR image. In an extreme example, the traditional discriminator won't penalize the generator for generating completely different but equally realistic images from an LR image. Although this is unlikely due to the existence of other content and perceptual losses, the gradient feedback given by the discriminator is sub-optimal for the task of super-resolution.

As a solution, we provide the low-resolution image as a reference along with the target image to the discriminator. This enables the discriminator to learn more important features for discriminating the generated image and provide better gradient feedback according to the LR image. For details, refer to Figure 1. We upsample the LR image to the same size as the HR image and concatenate them, feeding a tensor of shape $(H, W, 6)$ to the discriminator. Despite its simplicity, conditioning the discriminator on the input is a crucial modification for training such a supervised problem with GAN-oriented losses.

### 3.4 Blur detection

We recognized that there are often severely blurry regions in the images from the DIV2K[14] and DIV8K[15] datasets. Although the authors of [15] argue that the data was collected by "paying special attention to image quality", there were many scenes with out-of-focus backgrounds. These blurry regions might plague the generator to learn to generate such blurry patches. Blurry backgrounds are often indistinguishable from finer objects based only on the LR image. Though some might argue that the blurry backgrounds must also be learned, we were able to achieve finer detail and higher LPIPS score by detecting and removing blurry patches from both datasets.

We propose to detect and remove blurry patches before the network is trained on those patches. There are various methods for blur detection e.g. algorithmic methods and deep-learning-based approaches[11, 12]. However, most deep-learning-based works focus on predicting pixel-wise blur maps of the image, which wouldn't be suited for our needs. Mostly, the algorithmic method of [10] was successful at reliably detecting blurry patches as can be observed in Figure 3. We measure the variance of the Laplacian activation of the patch and consider patches with variance of under 100 as

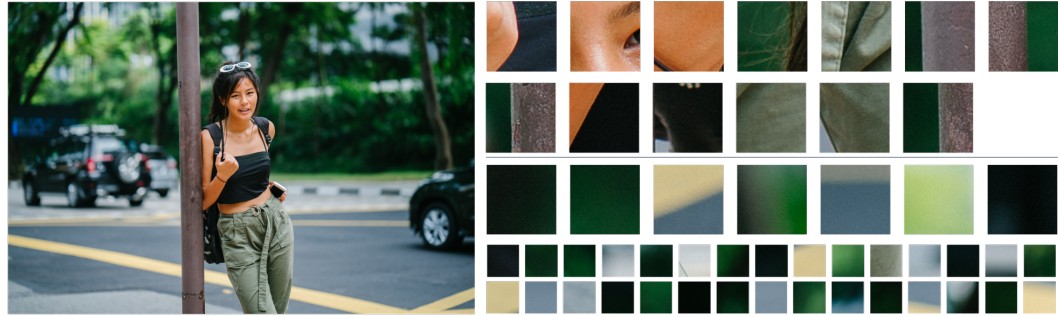

Figure 3: Randomly selected samples of the blur detection algorithm tested on image 0031 from the DIV8K dataset. The top two rows are the patches classified as clear and the bottom rows are blurry patches. Regions that are clear in the image (person, pole) are correctly considered as clear patches by the detection algorithm.

blurry patches. The algorithm detects 28.8% blurry patches in a sample of 16,000 randomly cropped patches of size 96×96 from the DIV2K dataset and 48.9% of patches in a sample of 140,000 patches from the DIV8K dataset.

# 4 Experiments

We conduct experiments to evaluate the effectiveness of our proposed techniques in ×4 and ×16 resolution and compare them with the baseline ESRGAN. We first experiment the effects of blur detection, then we perform an ablation study of our proposed training methods to evaluate their effectiveness. Implementation detail and training logs can be found on GitHub[1]. All our experiments were performed on a single Tesla T4 or Tesla K80 GPU on Google Colaboratory.

We observed that a large portion of the training was used for loading high-resolution images, despite most of the images not being used. As an implementation detail to improve training speed significantly, we extract multiple patches and save them in a buffer while training instead of extracting only a single patch after loading the image. We randomly pick images from the buffer for training and discard the selected patches from the buffer. In all of our experiments, we extract 128 patches from each image and create a buffer of 1024 patches.

## 4.1 ×4 super-resolution

### 4.1.1 Training details

We employ the ESRGAN network architecture with 23 RRDB blocks and most of its training configurations for the baseline of our experiments on ×4 super-resolution. The training process is divided into two stages. We first pretrain the PSNR-oriented models then train the ESRGAN-based models.

The PSNR-oriented models are trained with the L1 loss with a batch size of 16 for 500K iterations. We apply learning rate decay with an initial learning rate of $2 \times 10^{-4}$, decayed by a factor of 2 every $200k$ iterations. We initialize the GAN-based model with the PSNR-oriented model. We initialize the learning rate with $1 \times 10^{-4}$ for both $G_1$ and $D_1$, decaying the learning rate by a factor of 2 at [$50k$, $100k$, $200k$, $300k$] iterations. For optimization, we use the Adam optimizer for both pretrained networks and GAN-based models, with $\beta_1 = 0.9$ and $\beta_2 = 0.99$. The learning rate decay schedule corresponds to the one proposed by ESRGAN. We implement our models and methods with the Tensorflow framework. The loss function is scaled with $\eta = 10$ and $\lambda = 5 \times 10^{-3}$, which is equivalent to the training configuration of ESRGAN used in the PIRM-SR challenge. This is slightly different from the configuration used in the released model trained with $\eta = 10^{-2}$.

All of our networks are trained exclusively on the DIV2K dataset[14], while the original ESRGAN was trained with DIV2K, Flickr2K, and OST datasets combined. We obtained the LR images by

---

[1]`https://github.com/krenerd/ultimate-sr`

Table 1: LPIPS, PSNR, SSIM scores of various configurations for ×4.

| Methods | Set5 (LPIPS / PSNR / SSIM) | Set14 | BSD100 | Urban100 |
|---|---|---|---|---|
| Pretrained (a) | 0.1341 / 30.3603 / 0.8679 | 0.2223 / 26.7608 / 0.7525 | 0.2705 / 27.2264 / 0.7461 | 0.1761 / 24.8770 / 0.7764 |
| +Blur detection (b) | 0.1327 / 30.4582 / 0.7525 | 0.2229 / 26.8448 / 0.7547 | 0.2684 / 27.2545 / 0.7473 | 0.1744 / 25.0816 / 0.7821 |
| ESRGAN (Official) | 0.0597 / 28.4362 / 0.8145 | 0.1129 / 23.4729 / 0.6276 | 0.1285 / 23.3657 / 0.6108 | 0.1025 / 22.7912 / 0.7058 |
| ESRGAN (c) | 0.0538 / 27.9285 / 0.7968 | 0.1117 / 24.5264 / 0.6602 | 0.1256 / 24.6554 / 0.6447 | 0.1026 / 23.2829 / 0.7137 |
| +refGAN (d) | 0.0536 / 27.9871 / 0.8014 | 0.1157 / 24.4505 / 0.6611 | 0.1275 / 24.5896 / 0.6470 | 0.1027 / 23.0496 / 0.7103 |
| +Add noise (e) | **0.04998** / 28.23 / 0.8081 | 0.1104 / 24.48 / 0.6626 | **0.1209** / 24.8439 / 0.6577 | **0.1007** / 23.2204 / 0.7203 |
| +Cycle loss (f) | 0.0524 / 28.1322 / 0.8033 | **0.1082** / 24.5802 / 0.6634 | 0.1264 / 24.6180 / 0.6468 | 0.1015 / 23.1363 / 0.7103 |
| *-Perceptual loss* (g) | 0.2690 / 23.4608 / 0.6312 | 0.2727 / 22.2703 / 0.5685 | 0.2985 / 24.1648 / 0.5859 | 0.2411 / 20.8169 / 0.6244 |

downsampling the HR images with MATLAB bicubic interpolation. We compare the effects of our methods on LPIPS, PSNR, and SSIM scores on the Set5, Set14, BSD100, and Urban100 datasets. Scores evaluated on the Set5 and Set14 datasets are obtained by averaging the final 5 checkpoints, each recorded at [480*k*, 485*k*, 490*k*, 495*k*, 500*k*] iterations.

### 4.1.2   Ablation study

To study the effects of our proposed methods, we perform an ablation study of our proposed method. We enable our proposed methods one by one and list the resulting scores in Table 1. Each training configuration was fully trained with the original training configurations. We provide the saved model and configuration files to reproduce our results in our project repository. We also list the results of the official ESRGAN for fair comparison. The improvements from the official results and the result from configuration(c) is because the $\eta$ value is different from the official model. First, blur detection is experimented with in configuration(b) and improves the LPIPS score for all benchmarks. We train our baseline ESRGAN in configuration(c) and get reasonable results. By applying the technique of Section 3.3 in configuration(d), we slightly harm the network in terms of the LPIPS score. However, providing conditional information to the discriminator is crucial for learning such a supervised problem with adversarial learning. Our method of directly concatenating the reference image in the input is not optimal. The low-resolution image could be applied through SPADE[20] or alternative spatial transformation methods for improvements. Applying scaled noise shows large improvements as experimented in configuration(e).

The cycle consistency loss applied in configuration(f) shows neutral and slightly negative effects on the LPIPS score. The reason for this is mostly because of the incompetent GAN framework lacking the training techniques of modern GAN literature. Our statement is stated by the failure of configuration(g) where the GAN framework alone is responsible for learning the super-resolution process. The GAN framework of ESRGAN is incapable of lead the training process and thus the image quality wasn't improved when we gave more responsibility to the adversarial loss in configuration(f). However, coupled with improved GAN techniques in further research, the cycle consistency content loss will further enhance the image quality.

### 4.2   ×16 super-resolution

### 4.2.1   Training details

We employ the RFB-ESRGAN of [21] as the baseline for our experiments on ×16 super-resolution. The RFB-ESRGAN proposes an architecture using Receptive Field Blocks(RFB) and Residual of Receptive Field Dense Block(RRFDB), each as an alternative for convolution and RRDB blocks. The RFB-ESRGAN uses less memory compared to methods that manipulate the image in the intermediate ×4 resolution[22] and this allowed larger batch size in our environment. We employ the RFB-ESRGAN network architecture with 16 RRDB blocks and 8 RRFDB blocks for the baseline of our experiments on ×16 super-resolution.

The model is first trained with the L1 loss for 100K iterations with an initial learning rate $2 \times 10^{-4}$, decayed by a factor of two every $2.5 \times 10^5$ iteration. The GAN-based model is initialized with the pretrained model and is trained for 200K iterations, which is shorter than the original 400K iterations. Additionally, the batch size is decreased from 16 to 4 and we therefore approximately scale the initial learning rate of $10^{-4}$ to $2 \times 10^{-5}$ by a factor of 5. The learning rate is decayed at [50*k*, 100*k*]

Table 2: LPIPS, PSNR scores for various configurations for $\times 16$ super-resolution.

| Methods | DIV8K validation |
|---|---|
| Pretrained (a) | 0.4664 / 30.3603 |
| +Blur detection (b) | 0.4603 / 25.53 |
| RFB-ESRGAN(official) | 0.345 / 24.03 |
| Baseline RFB-ESRGAN (c) | 0.356 / 24.78 |
| Ours w/o cycle-loss (d) | **0.321** / 23.95 |
| Ours w/ cycle-loss (e) | 0.323 / 23.49 |

iterations. We don't use model ensemble to further stabilize the network. All other models and hyperparameter configurations are equal. We train the network on the DIV8K dataset[15], while the original network was trained with additional datasets including DIV2K, Flicker2K, OST dataset. The first 1,400 images of DIV8K are used as training data and the rest 100 validation images are used for evaluation.

### 4.2.2 Ablation study

The PSNR-oriented method is improved using blur detection in configuration(b). Our GAN-baed model of configuration(c) achieves worse performance compared the the results reported in [21] because of the lighter training configurations. We were able to make significant improvements in the LPIPS score from the baseline RFB-ESRGAN using our proposed methods in configuration(d). We apply all of our proposed methods except the cycle consistency loss in configuration(d). We also train the model with cycle consistency loss and get similar results in configuration(e). We were able to make such improvements using much lighter training configurations with only half iteration steps, $\times 4$ smaller batch size, and without model ensemble. The results are described in Table 2.

## 5 Conclusion

We proposed a one-to-many approach for super-resolution and achieve improved perceptual quality and better LPIPS score from the baseline ESRGAN configuration and achieve the state-of-art LPIPS score in x16 perceptual super-resolution. We provide scaled pixel-wise to the generator to allow stochastic variation in the reconstructed image and implement a generator capable of a one-to-many pipeline. We also address the limitations of mixing the strict content loss with perceptual losses and propose an alternative based on the cycle loss. Our newly modified loss will ensure the consistency of the content while not penalizing high-frequency detail. Additionally, we further propose more techniques such as blur detection using Laplacian activation and redesign the discriminator input by providing a reference image to further improve the perceptual quality of $\times 4$ and $\times 16$ super-resolution. However, the GAN framework from ESRGAN was incompetent to guide the training on its own. Modern GAN training techniques could be applied to further improve the GAN framework used in super-resolution. Our proposed loss function will become more effective as a content loss when coupled with a robust GAN framework since it will reduce constraints in generating high-frequency detail. Such improvements are left for future work.

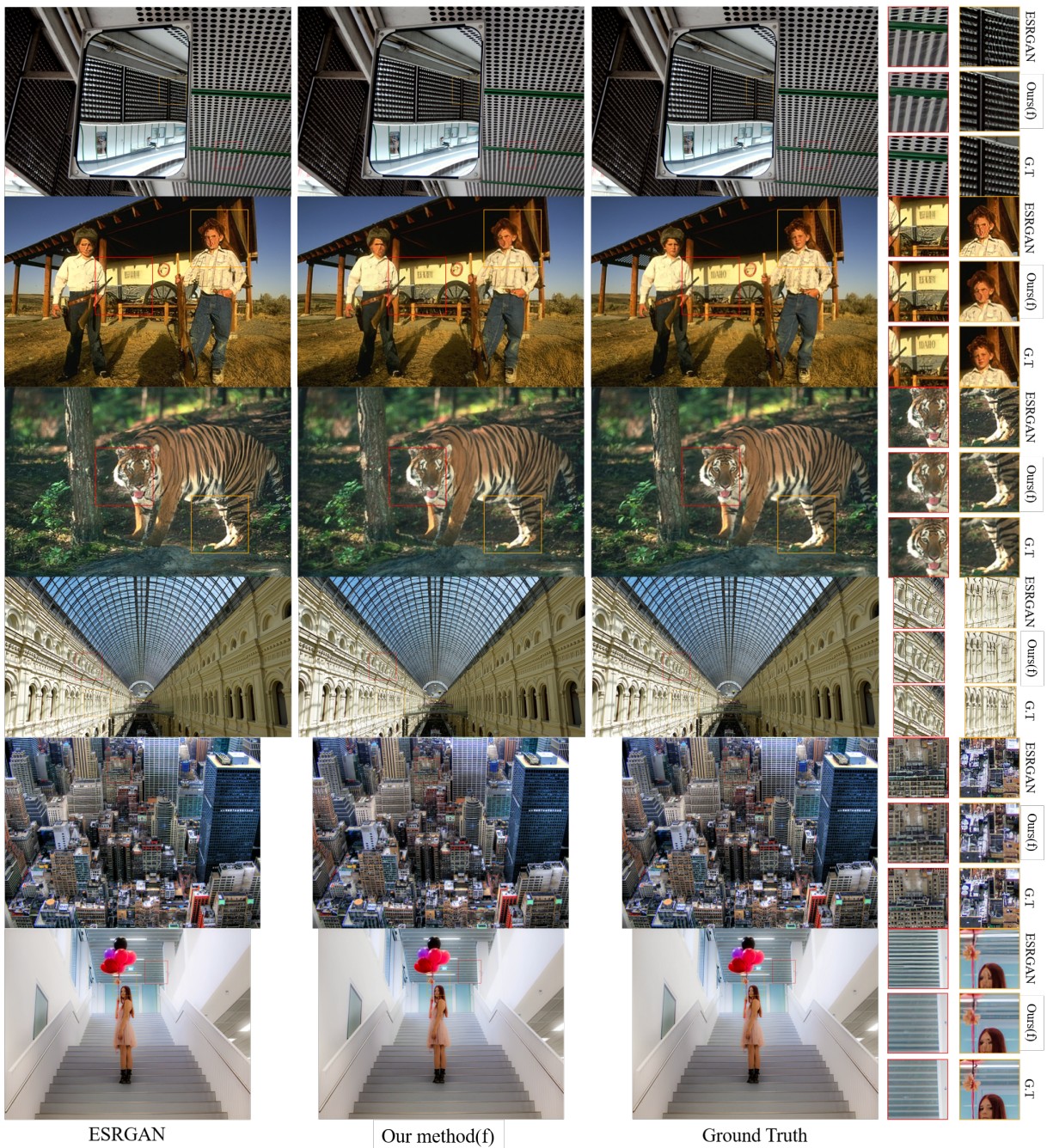

Figure 4: Qualitative comparison of our methods with the official ESRGAN. We compare the poorly reconstructed outputs of ESRGAN from BSD100 and Urban100 datasets with our proposed model trained with configuration(f). Our method produces sharp textures and more realistic structures compared to the baseline ESRGAN, although it also fails to accurately reconstruct human faces.

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
