# OpenReview forum: "One-to-many Approach for Improving Super-Resolution"
_NeurIPS.cc/2021/Conference — NeurIPS 2021 Submitted_

### Official Review · Reviewer_PHmW · 2021-07-13

**Rating:** 5
**Confidence:** 4

**Summary:**

This paper proposes to address the one-to-many problem of SR by adding random noise to each layer of generator and considering a new perceptual loss to allow high-frequency mismatch. The new network performs better than ESRGAN by further selecting non-blur area for training. Ablation studies verify the effectivness of new network design.

**Limitations And Societal Impact:**

This paper may consider adding the proposed techniques to other GAN-based methods to verify the effectiveness of various tricks. The new content cycle loss is interesting but up-sampling and down-sampling may induce some information loss in generation and discriminator. More comparisons to state-of-the-arts GAN-based methods is needed for drawing a persuasive conlusion for solving the one-to-may SR problem, since automatically selecting the most human-favarable estimation (without artifacts) from many possible estimations is still a hot research area and a non-trivial problem.

**Main Review:**

Pros:
- Noise generator to address one-to-many problem.
- New content loss not to penalize high-frequency gradients.
- Using laplacian to select non-blur area for training.
- Better subjective and objective quality than ESRGAN

Cons:
- Limited comparisons to state-of-the-arts, such as SRFlow and its improvments.
- Blured images compared with ESRGAN.


**Time Spent Reviewing:**

4

---

> ### Author Response · Authors · 2021-08-13
> **Response to reviewer PHmW**
>
> Thank you for providing us constructive feedback.
>
> Regarding the quote "Limited comparisons to state-of-the-arts, such as SRFlow[2] and its improvements.", we agree that our paper lacked comparisons with some key papers, including SRFlow mentioned by reviewers including you. We will upload an updated version of the paper regarding the comparisons with other S.O.T.A SR.
>
> Regarding the quote "Blurred images compared with ESRGAN.", we understood the sentence as that we didn't compare the effects of filtering blurred images on ESRGAN in Table 1. The results without blur detection are denoted as 'ESRGAN (Official)' and all other experiments were done with the filtering algorithm. Especially, the experiment on config (c) only modified the blur detection algorithm from the original ESRGAN. We agree that there was potential confusion in the layout, and have revised it.
>
> Regarding the comments related to "adding the proposed techniques to other GAN-based methods", GAN-based methods in super-resolution are not diverse. They stem from the work of SRGAN and typically use a combination of content, perceptual(VGG score), and GAN loss as the training objective of the generator. Most modifications are made to the network architecture and modifications to the training objectives are minor. Recent works on GAN-based SR is typically based on the ESRGAN pipeline, and we believe that the effects of the techniques in this paper can sufficiently generalize to other GAN-based methods in SR. The results of the experiments on x16 SR using RFB-ESRGAN can be an example.
> Also, we note that SRFlow uses a very different approach from GAN-based methods and most of our proposed techniques are inapplicable to SRFlow.
>
> Regarding the quote "The new content cycle loss is interesting but up-sampling and down-sampling may induce some information loss in generation and discriminator.", this 'weak-content loss' is actually the basis of our work. We supposed that the generation of high-frequency, photo-realistic details should be done purely on the perceptual and GAN loss. The role of the content loss is to simply make sure that the reconstructed image is consistent with the original image, and thus won't interfere with other objectives being 'creative' on generating pixel-level differences. We believe that our proposed content loss based on the cycle-consistency loss is the optimal form for ensuring consistency while not interfering with other perceptual objectives.
> In short, our work claims that the loss of information in the content loss is actually the key to relying more on perceptual objectives. We also added some clarification on the paper on why we adopted a weaker content loss.
>
> Thank you again for your generous feedback.
>
> [1] Wang, X., Yu, K., Wu, S., Gu, J., Liu, Y., Dong, C., ... & Change Loy, C. (2018). Esrgan: Enhanced super-resolution generative adversarial networks. In Proceedings of the European conference on computer vision (ECCV) workshops (pp. 0-0).
>
> [2] Lugmayr, A., Danelljan, M., Van Gool, L., & Timofte, R. (2020, August). Srflow: Learning the super-resolution space with normalizing flow. In European Conference on Computer Vision (pp. 715-732). Springer, Cham.

---

### Official Review · Reviewer_44pW · 2021-07-16

**Rating:** 4
**Confidence:** 5

**Summary:**

This paper focuses on addressing the ill-posed problem in super-resolution task. To this end, the authors propose a new data augmentation with Laplacian activation, soft content loss and adding noise into the extracted features. The experimental results show the proposed method can superior the baseline ESRGAN on Set5, Set14, BSD100 and Urban100.

**Ethics Review Area:**

["I don’t know"]

**Limitations And Societal Impact:**

1. Important references are mssing, such as:
[1] Amortised MAP Inference for Image Super-resolution
And the author cannot claim to be the first to consider SR as a one-to-many problem.
2. The author should compare more recent state-of-the-art methods in Table 1.
3. Although the adding scaled Gaussian noise to the generator appears to be novel and brings some marginal improvements on performances, its motivation is weird to me. The submission says “the sensitivity and the desired magnitude of noise would differ for each channel.” However, it is not intuitive. Do you have any further justification of this assumption?
4. Overall, I feel the three contributions are not well justified/evaluated

**Main Review:**

Strengths
1. The idea of the paper is clearly written and empirical results demonstrate its effectiveness.
2. The paper presented many interesting ideas, e.g., soft content loss, new data augmentation with blur detection, one-to-many generator, and conduct extensive ablation study for each of the terms.
3. There are additional results in Table 2 of the paper that validate the key idea for x16 SR.

Weaknesses
1. The authors cannot claim this paper as the first to consider SR as a one-to-many problem.
Actually many previous papers have already model SR in a one-to-many way. For example, [1] gave a solid theoretical explaination of SR problems and give an effective solution. And GAN based SR methods, use discriminator loss for the same reason.

[1] Amortised MAP Inference for Image Super-resolution

2. The motivation of adding noise to intermediate layers seems lacking theoretical support.
Indeed, adding noise can generate various outputs. However, candidate multiple HR images are not just random. They lie in some `null` space  of the downsample operator. And adding random noise cannot ensure to generate proper HR results.

3. The major weakness is the novelty of the proposed method, although the ablation studies validate the effectives of each technique on several datasets. The novelty is limited, and there are few new techniques for me. In my view, the proposed method seems to stack many tricks for improving the performances, such as a new data augmentation and adding the pixel-wise noise into the extracted features.

4. Table 1, The comparison to STDA methods do not include the recent state of the art methods like " Enhancing Perceptual Loss with Adversarial Feature

5. Matching for Super-Resolution", IJCNN’20; “Journey Towards Tiny Perceptual

6. Super-Resolution”, ECCV’20; “Generative adversarial network-based image super-resolution using perceptual content losses”, ECCV’18 (and many others).

7. The manuscript is filled with informal words and needs a rewrite.
28: don’t -> do not
98: “->” -> “\rightarrow”
125: won’t -> will not
224: don’t -> do not
241: “x16 “ -> “\times 16”




**Time Spent Reviewing:**

1

---

> ### Author Response · Authors · 2021-08-18
> **Response to reviewer 44pW**
>
> Thank you for your constructive feedback.
>
> After reviewing more references suggested by reviewer 44pW and the SRFlow paper pointed out by other reviewers, we decided to change the objective of our paper from "proposing a one-to-many approach for SR" to "adapting the concepts of SRFlow to improve GAN-based SR by properly implementing the one-to-many property" since our method did improve the perceptual quality of GAN-based SR methods(ESRGAN, RFB-ESRGAN).
>
> Although our work doesn't seem to meet the standards of the submitted to NeuralIPS, we thank all reviewers for providing great constructive feedback and enabled us to improve our work and learn.

---

### Official Review · Reviewer_rx3L · 2021-07-16

**Rating:** 3
**Confidence:** 5

**Summary:**

This paper develops new loss functions, network architectures, and training methods for solving the single image super resolution problem using convolutional neural networks. The aim of this effort is to develop a network that is capable of producing the many plausible HR images that could have produced a LR image. With regards to loss functions, the authors propose penalizing the difference between a downsampled HR reconstruction and the input LR image (rather than a GT HR image) and also conditioning a discriminator network on the low resolution image. With regards to network architectures, the authors pump noise into the intermediate layers of the network to generate diverse outputs. With regards to training methods, the authors remove smooth patches from the DIV2K dataset for training.

**Limitations And Societal Impact:**

At 16x, super-resolution is essentially hallucinating plausible scenes. It's worth adding a sentence that even if a super-res algorithm could produce pleasing reconstructions, they don't necessarily represent the truth.


**Main Review:**


# Originality
Other recent works have noted that there should be multiple solutions to the SISR problem and developed their own solutions [A]. These methods are not mentioned nor compared against.

The others innovations in this work don't seem particularly novel.

[A] Lugmayr, Andreas, et al. "Srflow: Learning the super-resolution space with normalizing flow." European Conference on Computer Vision. Springer, Cham, 2020.


# Quality
The paper provides few high quality images and those that are included aren't presented clearly: Is Figure 4 demonstrating 4x or 16x superresoltuion results? I think 16x because both methods have major artifacts.

The comparisons are each limited to a single method that is no longer SOTA. See [A].

Much of the paper is motivated by the idea the one LR image should produce multiple plausible HR images. However, the main text provides no examples of this occuring.

User studies are preferable to relying on LPIPS alone.

# Clarity
The authors motivate the L1 penalty in the LR domain as part of a cycle consistency loss. Howevever, cycle consistency seems fundamentally at odds with the goals of the paper; generating a diverse set of HR outputs form a LR input. A true cycle consistent loss would ensure that the LR image formed by a single HR image would always produce that same HR image.

# Significance
The paper set out to develop a one-to-many approach to super-resolution, which is arguably an important problem. Unfortunately, there's little evidence this was accomplished.


**Time Spent Reviewing:**

1.5

---

> ### Author Response · Authors · 2021-08-18
> **Response to Reviewer rx3L**
>
>
> After reviewing more references including the SRFlow paper pointed out by reviewer rx3L and other reviewers, we decided to change the objective of our paper from "proposing a one-to-many approach for SR" to "adapting the concepts of SRFlow to improve GAN-based SR by properly implementing the one-to-many property" since our method did improve the perceptual quality of GAN-based SR methods(ESRGAN, RFB-ESRGAN).
>
> We did want to point out that our work does share the same conceptual ideas with SRFLow, but is practically very different from SRFlow. There seem to be misunderstandings about our cycle-consistency loss. Conceptually, we train the generator to generate high-frequency details with the GAN and perceptual loss, while our content loss simply ensures the reconstructed image to be consistent with the original image. Our content loss will penalize the various images while the semantic content is consistent with the low-resolution image.
>
> Also, with regard to your comments related to user studies, the 2020 NTIRE challenges suggest that LPIPS scores are highly consistent with user studies on evaluating the perceptual quality of super-resolved images.
>
> Although our work doesn't seem to meet the standards of the submitted to NeuralIPS, we thank all reviewers for providing great constructive feedback and enabled us to improve our work and learn. We had a great experience throughout the review process and are hoping to participate in later NeurlIPS conferences with a better paper. We thank reviewers again for constructive feedback.

---

### Decision · Program_Chairs · 2021-09-27

**Decision:**

Reject

**Comment:**

The paper proposes a one-to-many approach to super-resolution. All reviewers and the AC agree that this is an important problem and a very interesting research direction.

The original manuscript missed several important prior work (as listed by Reviewers rx3L and 44pW), which also treat SR as a one-to-many problem. This significantly reduced the novelty aspect of the work. In their rebuttal the authors acknowledged these similarities and proposed to change the objective of work to "adapting the concepts of SRFlow to improve GAN-based SR by properly implementing the one-to-many property". The AC agrees with the authors that the paper does improve performance and make interesting contributions (see below). However, incorporating these changes, adding new baselines and reformulating the work would certainly require another round of revisions.

The paper makes several practical contributions (soft content loss, pixel-wise noise after each RRDB block, filter blurry regions using Laplacian activation) that seem to work well in practice. This was highlighted by Reviewer 44pW.

Reviewer rx3L points out that the paper does not clearly show the ability of the model of generating multiple HR images from a given LR image. The AC agrees that this would strengthen the work significantly.

The AC agrees with the concerns raised by the reviewers and finds no basis for overruling their recommendation. The AC appreciates the constructive rebuttal and encourages the authors to incorporate the reviewers feedback and resubmit to a future conference.